# Characterization of the DNA accessibility of chloroplast genomes in grasses
Yinmeng Liu[1,2], Jinling Zhan[1,2], Junjie Li[1,2], Mengjie Lian[1,2], Jiacheng Li[1,2], Chunjiao Xia[3], Fei Zhou[1,3] & Weibo Xie [1,2] ✉

Although the chloroplast genome (cpDNA) of higher plants is known to exist as a large protein-DNA complex called 'plastid nucleoid', researches on its DNA state and regulatory elements are limited. In this study, we performed the assay for transposase-accessible chromatin sequencing (ATAC-seq) on five common tissues across five grasses, and found that the accessibility of different regions in cpDNA varied widely, with the transcribed regions being highly accessible and accessibility patterns around gene start and end sites varying depending on the level of gene expression. Further analysis identified a total of 3970 putative protein binding footprints on cpDNAs of five grasses. These footprints were enriched in intergenic regions and co-localized with known functional elements. Footprints and their flanking accessibility varied dynamically among tissues. Cross-species analysis showed that footprints in coding regions tended to overlap non-degenerate sites and contain a high proportion of highly conserved sites, indicating that they are subject to evolutionary constraints. Taken together, our results suggest that the accessibility of cpDNA has biological implications and provide new insights into the transcriptional regulation of chloroplasts.

The chloroplast genome (cpDNA) is typically between 0.1–0.3 Mb in size and contains ~100–200 genes encoding 50–200 proteins[1]. A cpDNA is usually displayed as a circular molecule with a quadripartite structure consisting of large single copy region (LSC), small single copy region (SSC), inverted repeat region A (IRA), and inverted repeat region B (IRB)[2].

Studies based on microscopy and live-imaging have shown that cpDNA has a multibranched linear structure in many angiosperm species, and multiple copies of cpDNA are organized into DNA–protein complexes called "chloroplast nucleoids"[3]. Proteomic studies have shown that the nucleoids contain diverse DNA-binding proteins[4], of which SiR is a major component[5,6]. SiR can induce reversible compression of chloroplast nucleoids, thus inhibiting transcription and replication. Similarly, WHIRLY1 is a major protein of chloroplast nucleoids, and microscopic analyses showed that the WHIRLY1 protein compacts the DNA of a subpopulation of plastid nucleoids[7]. Monokaryotic chloroplast 1 (MOC1) was reported to maintain chloroplast nucleoid structure by resolving Holliday junctions[8]. Overall, these studies suggest that the structure of cpDNA is not static but dynamically bound by different proteins at different stages[9,10].

In contrast to the studies of the chloroplast proteome, researches on the binding sites of these proteins on cpDNA and the DNA state of cpDNA (e.g., binding state of transcription factors or other proteins and the DNA accessibility) have been relatively limited[2]. New techniques have emerged in

recent years to rapidly obtain information on the chromatin state and regulatory elements of eukaryotic nuclear genomes. The ATAC-seq (assay for transposase-accessible chromatin sequencing), which uses a modified Tn5 transposase to insert sequencing adapters into accessible chromatin regions, provides an efficient way to obtain genome-wide accessible chromatin information[11,12]. Meanwhile, the binding of transcription factors provides protection for DNA, resulting in a dramatic reduction of Tn5 insertions at the binding sites, forming 'footprints'. By comparing the distribution of Tn5 insertions, one can identify such footprints, which indicate potential regulatory elements[13]. Besides the nuclear genome, recent studies found that the ATAC-seq technique can also be used to study the DNA state of mitochondria (mtDNA) in mammals and observed that the accessibility and footprinting patterns of the mtDNA are dynamically altered during embryogenesis[14,15]. These studies encouraged us to consider whether this approach could be extended to investigate the DNA state of the cpDNA in grasses. If so, what are the accessibility patterns of cpDNA in grasses? Do accessibility patterns have biological significance for transcriptional regulation in chloroplasts?

In this study, by generating ATAC-seq data for multiple tissues of five grasses, we constructed a high-resolution accessibility map of cpDNA and characterized the accessibility of cpDNA of five grasses. Our results suggest that the accessibility of cpDNA is associated with transcriptional regulation

[1]National Key Laboratory of Crop Genetic Improvement, Hubei Hongshan Laboratory, Huazhong Agricultural University, Wuhan 430000, China. [2]Hubei Key Laboratory of Agricultural Bioinformatics, College of Informatics, Huazhong Agricultural University, Wuhan 430000, China. [3]College of Life Science and Technology, Huazhong Agricultural University, Wuhan 430000, China. ✉e-mail: weibo.xie@mail.hzau.edu.cn

of chloroplast genes. We further identified footprints in cpDNA and found that they are co-localized with known functional elements, and their distribution and sequences are subject to selection. Our results suggest that the accessibility of cpDNA has biological implications and provides new insights and resources for understanding the transcriptional regulation of chloroplasts.

## Results

### An accessibility map of the chloroplast genome in grasses

We generated 55 high-quality datasets (53 for ATAC-seq and 2 for RNA-seq, Supplementary Tables 1, 2) from five tissues (root, young leaf, flag leaf, young panicle, and spikelet) of five species (*O. sativa, B. distachyon, S. italica, S. bicolor,* and *Z. mays*) that were initially planned to be used to study the regulation of the nuclear genomes in grasses. Since organelles cannot be completely removed during library construction for ATAC-seq and RNA-seq, these data can also be used to study the chloroplast genomes. To explore the accessibility characteristics of cpDNA in grasses, we specifically developed a workflow (Fig. 1a). After mapping the sequencing data to cpDNA and merging replicates, 80% of the ATAC-seq datasets had more than 1,000,000 matched reads, corresponding to more than 1000-fold coverage of cpDNA. In addition, RNA-seq datasets had an average of more than 250,000 reads for a single tissue in rice, equivalent to more than 250-fold coverage of cpDNA (Supplementary Tables 1–3), suggesting sufficient data for subsequent analysis.

To verify the differences in chromatin structure between cpDNA and nDNA, we looked at the size distribution of ATAC-seq library fragments matched to cpDNA and nDNA, respectively. The distribution of fragment sizes reflected that the chloroplast genome, indeed, did not have the nucleosome pattern observed in the nuclear genome (Supplementary Fig. 1a–c).

By dividing cpDNA into 100-bp bins and quantifying the number of Tn5 insertions per bin as accessibility, we calculated the correlations of accessibility between different datasets. We observed high correlation between replicates and the clustering of tissues consistent with biological features in the five species, indicating that the accessibility of cpDNA has biological relevance (Supplementary Fig. 1d).

By calculating the read coverage of RNA-seq and Tn5 insertions of ATAC-seq across multiple tissues separately, we constructed a comprehensive map of gene expression and accessibility in cpDNA of grasses, which is illustrated in Fig.1b for rice. As expected, the map showed that the entire cpDNA was accessible. However, we noticed that the accessibility of different regions in cpDNA varied widely and the most accessible regions in cpDNA were the reverse repeat regions (IRA and IRB), which also exhibited the highest expression signals. These clues prompted us to hypothesize that the expression level of chloroplast genes may be related to their accessibility.

### Altered accessibility of the chloroplast genome is linked to gene expression and regulation

To characterize the accessibility distribution of cpDNA, we first divided cpDNA into genic and intergenic regions, and quantified Tn5 insertions. The Mann–Whitney *U*-test indicated that the accessibility of the intergenic region was markedly lower than the genic region (*P* value <1.0e-7, Fig. 2a). We further divided cpDNA into 100-bp bins and quantified both the RNA-seq coverage and Tn5 insertions for each bin. We then grouped the bins according to the expression level and accessibility, respectively, and calculated the overlap of bins between groups (Fig. 2b). We clearly observed that highly accessible regions of cpDNA tended to overlap with regions that had higher expression signals, while low accessible regions tended to have lower expression signals. The 20% bins with the highest expression signal accounted for 50% of the 20% regions with the highest accessibility, 89.9% of which were annotated as genic regions. The 20% bins with the lowest expression signal accounted for 40% of the 20% regions with the lowest accessibility, 80.4% of which were annotated as intergenic regions (Fig. 2b). Furthermore, we performed the same analysis on the nuclear genome and found that the correlation between the expression signal and accessibility was more pronounced in cpDNA (Fig. 2c). Together, these findings

support that the accessibility of cpDNA is strongly related to the level of gene expression.

In nDNA, the Tn5 insertion in ATAC-seq is enriched at the gene start site (GSS) because the transcription factors are enriched at the promoter expelling the nucleosomes, resulting in open chromatin regions[16]. Moreover, the promoter regions of the highly expressed genes usually have higher accessibility than the lowly expressed ones in nDNA[17]. What would be the case since cpDNA has no nucleosomes? We calculated Tn5 signals near the GSSs and gene end sites (GESs) (±1 kb) for highly expressed genes (top 30%) versus lowly expressed genes (bottom 50%) in cpDNA in rice young leaves and roots, respectively. In contrast to the results in nDNA, in both tissues, we observed that genes with higher expression exhibited deeper depletion of Tn5 signal at the upstream of GSSs in both tissues, suggesting more protein binding events (Fig. 2d). Moreover, higher Tn5 signals were detected in the bodies of highly expressed genes than lowly expressed genes, consistent with our observations obtained in Fig. 2b that the accessibility of gene bodies in cpDNA is positively correlated to the level of gene expression. In addition, genes with lower expression exhibited deeper depletion of Tn5 signals at the downstream of GESs (Fig. 2e), possibly reflecting that proteins might bind here to repress the transcription, like the transcription termination factors reported in human mtDNA and *Arabidopsis* cpDNA[18–21]. In the above analysis, we did not consider the structure of the operons in cpDNA, as studies in barley have shown that most genes within the operons also have promoters[22]. To further confirm the results, we also collected 43 reported operons[23,24] and performed the same analyses considering only their start and end sites in rice cpDNA as GSS and GES, and observed similar results (Supplementary Fig. 2).

Taken together, our results suggest that the variation of accessibility in different regions reflects the state of gene expression and regulation in cpDNA.

### Identifying accessibility footprints of chloroplast genomes in five grasses

We further conducted footprinting analysis to screen the potential regulatory elements in cpDNA across all five grasses. In general, 53 ATAC-seq datasets were used to identify accessibility footprints of cpDNA (CPAFPs) by pyDNase[25]. After merging the results from different tissues, an average of 794 high-confidence CPAFPs (FDR <0.01) were identified per species, with 648 for *O.sativa*, 578 for *B.distachyon*, 626 for *S.italica*, 1038 for *S.bicolor*, and 1080 for *Z.mays* (Supplemental Data 1–5 and Supplementary Table 3).

One issue that may affect the identification of CPAFPs is the insertion bias of Tn5. Although the Tn5 enzyme has been modified and optimized, the sequence preference for its insertion sites still exists[26]. We, therefore, used Tn5 tagmentation data of naked genomic DNA (gDNA) to correct for the effects of Tn5 bias on the identification of footprints (Methods). We scored every footprint using footprint depth (FPD, Fig. 3a) scores before and after the correction. Footprints with high FPD scores were more likely to be occupied by proteins. The results showed that despite the overall decrease after correction, the FPD scores before and after correction were highly correlated, implying a moderate effect of Tn5 bias on CPAFP identification, but a small effect on most high-scoring FPDs (Fig. 3b, *R* = 0.67, *P* value <1e-17). We plotted the accessibility patterns of CPAFPs and the flanking 100-bp. The results showed that the Tn5 signal in the center of CPAFPs was remarkably reduced in both ATAC-seq and gDNA data, while the corrected Tn5 signal still had a similar pattern (Fig. 3c). We also inspected the Tn5 signal around CPAFPs that overlap with known functional elements. The results showed that CPAFPs overlapping with the known replication initiation site oriA remained convincing after correction (Fig. 3d, gray area). Taken together, these results provide preliminary evidence for the biological significance and validity of CPAFPs. Considering that some CPAFPs may be associated with DNA modifications, and the naked gDNA retains the DNA modifications, so the corrected signal may lose much valid information[27], and the gDNA data of the other four species are not available, so the subsequent analyses are still based on the pyDNase results of raw Tn5 signals.

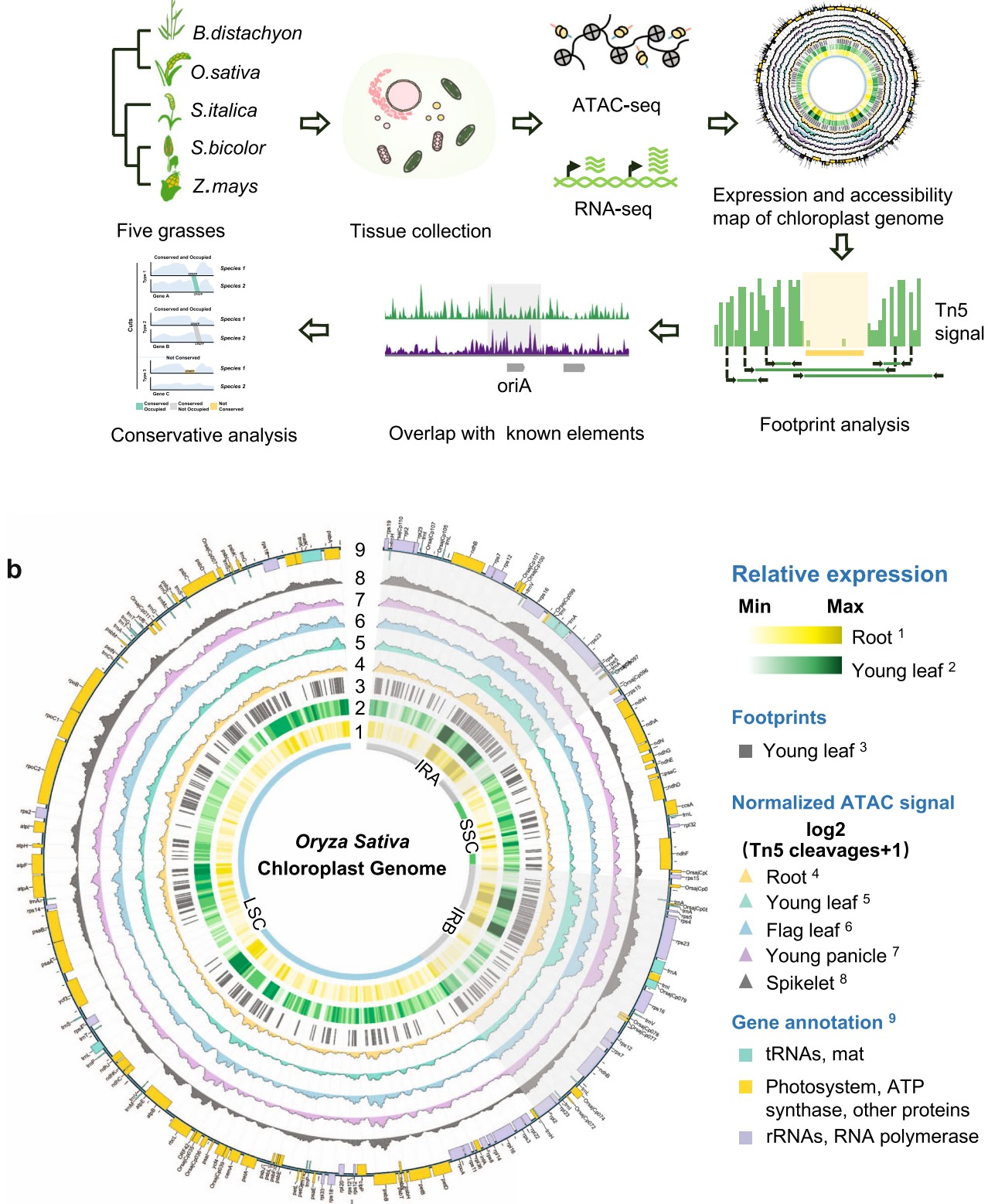

**Fig. 1 | The accessibility map of the chloroplast genome in rice. a** The cpDNA accessibility analysis workflow. **b** The circular plots of the accessibility and expression map. The tracks from the inside represent the quadripartite structure of cpDNA, the IRA and IRB regions were covered by the gray sector ring; gene expression signal in root[1] and young leaf[2]; footprints identified in young leaf[3]; accessibility signal of root[4], young leaf[5], flag leaf[6], young panicle[7], and spikelet[8]; annotated gene features on two strands[9].

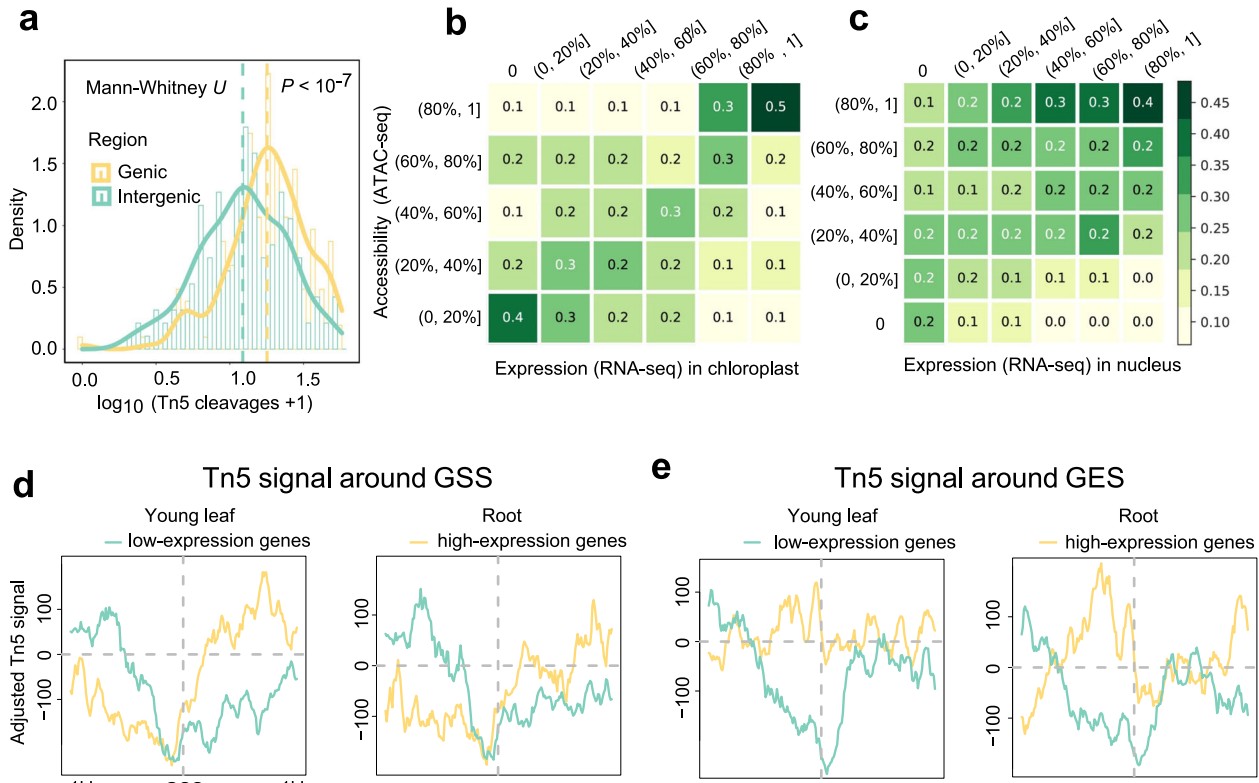

**Fig. 2 | The variation of accessibility in different regions and its relationship to expression. a** The distributions of accessibility in genic and intergenic regions. The x-axis is the average Tn5 cleavages per region (first sum the Tn5 cleavages for each genic and intergenic regions, then divided by its length), while the y-axis represents the density. Note that the structures of the operons are not considered in this analysis. The analysis was conducted using ATAC-seq data from rice young leaves after combining replicates. If not specifically mentioned, the analyses in the subsequent sections are all based on this data. **b** The distribution of accessibility of regions with different expression signals in cpDNA. In the heatmap, the columns represent different levels of RNA-seq coverage and the rows represent different levels of Tn5 signal of ATAC-seq. The numbers in the cell represent the proportion of regions with a certain level of expression that have a certain level of accessibility, for example, the upper-right cell indicates that 50% of the top 20% high-expression regions have their accessibility at the top 20% level. **c** The distribution of accessibility of regions with different expression signals in the nuclear genome. The ATAC-seq data used, and the meaning of the heatmap are the same as (**b**). **d**, **e** Accessibility around the gene start (GSS) and end sites (GES) in cpDNA. Annotated genes in cpDNA were classified as "low-expression" (bottom 50%) and "high-expression" (top 30%) classes according to the expression levels. The average accessibility of 200 bp at both ends (400 bp in total) was defined as the background and subtracted from the average accessibility at each position for adjustment.

Another issue that may affect our study is the reads that mapped to cpDNA may be contaminated by DNA fragments that were transferred from cpDNA into the nuclear genome during evolution, known as nuclear plastid DNAs (NUPTs)[28]. To assess the influence of NUPTs on the accessibility and footprinting analysis, we estimated the proportion of NUPT reads accounted for all the reads mapped to cpDNA in both gDNA tagmentation data and ATAC-seq data (Methods). The results showed that the averages of this proportion in the gDNA and ATAC-seq data of rice young leaves were about 2 and 0.2%, respectively (Supplementary Fig. 3a). We also calculated this proportion for all ATAC-seq datasets, which was below 2% for all datasets except spikelets, implying that NUPT reads made up a rather low proportion of all the reads mapped to cpDNA and had only negligible impact on the identification of CPAFPs (Supplementary Fig. 3b).

### CPAFPs colocalize with known functional elements in the chloroplast genome

We further wondered whether the identified CPAFPs overlap with known functional elements. PEP promoters typically contain two conserved elements, which are usually located in the 10 bp (TATAAT) and 35 bp (TTGACA) upstream of the transcription start site and are essential for the transcription of chloroplast genes (Fig. 3e)[22]. Previous study[29] in *Arabidopsis* reported cpDNA genes with promoters containing these two elements, and we confirmed that they were conserved between *Arabidopsis* and rice by sequence alignment (Fig. 3e). The Tn5 signal around the TSS of these genes

showed, as expected, a clear depletion of the Tn5 insertions near the -10 and -35 sites, indicating that they are likely protected by protein binding (Fig. 3e). The decline in accessibility is not as pronounced in the -35 region as in the -10 region. A possible explanation is that the -10 element is more conserved compared to the -35 element and plays a more important role in the transcription of chloroplast genes[22], therefore it might be bound more frequently.

Another known element in cpDNA is the origins of replication (ori), which has been reported to be bound by proteins[30]. We obtained the sequences of ori in the previous study[31,32]. The two copies of oriA were found to be highly conserved among the five grasses used in this study, while oriB was not conserved, and CPAFPs identified in multiple tissues of rice were overlapped with both oriA copies (Fig. 3f). We also looked at CPAFPs identified in other grasses and found that overlapping CPAFPs were detected on oriA in almost all other grasses (except for one ori site in *S.italica*, but it also exhibited similar footprint patterns), further supporting the biological importance of CPAFPs (Fig. 3g and Supplementary Fig. 4).

In summary, CPAFPs co-localized with known functional elements, suggesting that CPAFPs may provide a reference for exploring new regulatory elements in cpDNA.

### The distribution of CPAFPs in CDS regions is subject to selection

As mentioned earlier, intergenic regions are less accessible than genic regions, we speculated that many regulatory elements located in the

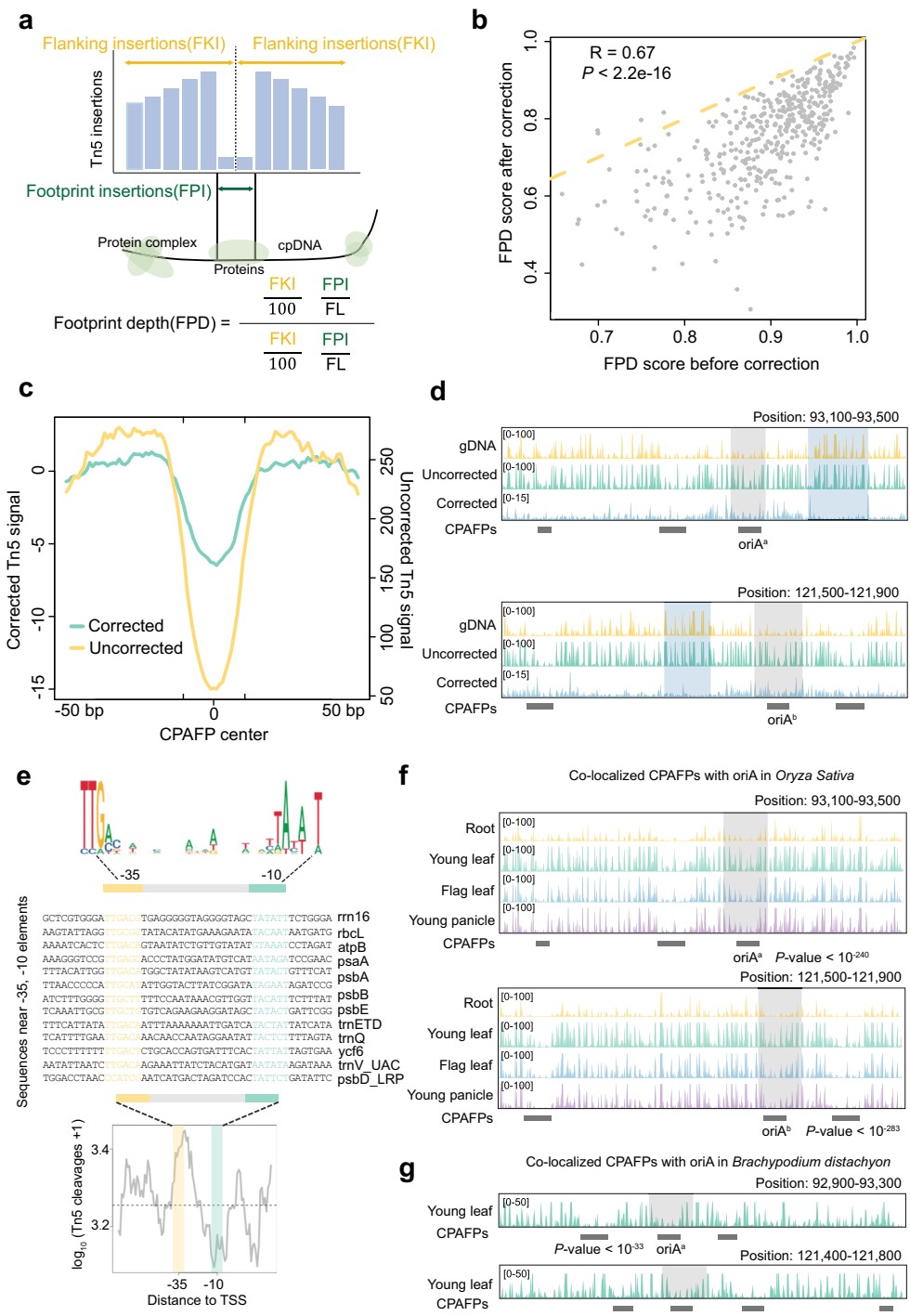

**Fig. 3 | Analysis of accessibility footprints of cpDNA (CPAFPs). a** Diagrammatic illustrating the calculation of footprint depth (FPD). FPI: the number of Tn5 insertions in the footprint region. FKI: the number of Tn5 insertions in the flanking 50-bp region from the center (100 bp total). FL: the length of the footprint. **b** The correlation of FPD scores before and after correction using Tn5 tagmentation data of naked genomic DNA (gDNA). Each point represents a CPAFP ($n = 413$) with FDR <0.01. **c** Patterns of Tn5 insertions around CPAFPs. The average signal of flanking 50 bp from the CPAFP center (100 bp in total) was defined as the background and subtracted from the signal at each position for adjustment. **d** Profiles of Tn5 insertions per site near the oriA site in rice before and after correction. The gray stripe marks the oriA site. **e** The upper panel is the sequence logos for the -35 and -10 elements of PEP promoters reported in *Arabidopsis*[29]. The bottom panel is the pattern of Tn5 insertions around the -35 and -10 elements of PEP promoters in rice young leaves. **f, g** Profiles of Tn5 insertions around the two replication initiation sites, oriA, in multiple tissues of rice (**f**) and in young leaves of *B. distachyon* (**g**). The *P* values of CPAFPs overlapping with oriA in young leaves of rice are indicated.

**Table 1 | Fisher's exact test of the CPAFPs distribution preference**

|  | Non-overlap with CPAFP | Overlap with CPAFP | Total |
|---|---|---|---|
| Genic* | 98,669 | 6896 | 105,565 |
| Intergenic | 26,634 | 2326 | 28,690 |
| Total | 125,303 | 9222 | 134,525 |
|  | **OR = 1.25; P value <2.2e-16** | | |

*Defined by the boundaries of the 43 reported operons[22].

**Table 2 | Fisher's exact test of the 4DS distribution in CPAFPs**

|  | Non-overlap with CPAFP | Overlap with CPAFP | Total |
|---|---|---|---|
| 4DS | 8894 | 646 | 9540 |
| Non-4DS | 16,431 | 2296 | 18,727 |
| Total | 25,325 | 2942 | 28,267 |
|  |  | **OR = 1.92; P value < 2.2e-16** | |

intergenic regions, thus they may be protected from the Tn5 cleavage by forming "protein–DNA" structures. To check the hypothesis, we defined genetic and intergenic regions based on the 43 reported operons[22] and analyzed the CPAFP distribution preference between genic and intergenic regions. The results showed that although intergenic regions covered 21.3% (28,690 bp) of the cpDNA, it comprised 32.0% (132 for 413) of footprints, indicating that the footprints significantly prefer to be located in the intergenic regions (Fisher's exact test, P value <2.2e-16, odds ratio = 1.25, Table 1).

The genic regions, mainly the coding regions, also have a large number of footprints. Recent studies revealed that there are some regulatory elements located in coding regions that play a dual role in determining gene expression in addition to amino acid sequences, termed duons[33]. If these footprints we identified belong to duon and are biologically meaningful, they may be subject to selection and thus show certain characteristics in their distribution. We analyzed the overlap of CPAFPs with fourfold degenerate sites (4DSs) in CDS regions, which are generally considered to be less constrained by selection. A total of 9540 4DSs were identified in rice cpDNA (Methods), of which 6.8% of the 4DSs (646/9,540) overlapped with CPAFPs, while 12.3% of the non-4DSs (2296/18,727) overlapped with CPAFPs, indicating that CPAFPs tend to overlap with non-4DSs (Fisher's exact test, P value <2.2e-16, odds ratio = 1.92, Table 2), which suggests that CPAFPs may tend to couple to coding sequences for more stable inheritance.

Since we collected data from multiple grasses, this prompted us to analyze how this characteristic has changed over the evolution. By mapping CPAFPs identified in other grasses to rice cpDNA (Methods), we identified 108 CPAFPs in genic regions that are both conserved in sequence and identified as footprints in all five grasses (Supplemental Data 6). We found that they contained only 0.8% of 4DSs (81/9540) but 4.4% of non-4DSs (819/18,727), showing that the conserved footprints tend to overlap with non-4DSs (Fisher's exact test, P value <2.2e-16, odds ratio = 5.34, Supplementary Table 4), which further suggests that the distribution of CPAFPs in CDS regions is subject to selection during evolution.

**Dynamics of accessibility and CPAFPs between leaves and roots**
In rice, 27% percent of CPAFPs were detected in all five tissues (Supplemental Data 1), while 28% of CPAFPs were identified in only one tissue, suggesting their roles in dynamic regulation. We next asked whether the dynamic changes in CPAFPs between different tissues were biologically meaningful. We observed that the accessibility of flanking 50-bp regions (total 100 bp, same below) and the FPD scores of most CPAFPs were generally similar among different tissues (Supplementary Fig. 5b). However, after classifying CPAFPs in genic regions according to the function of their host genes, we noticed that the accessibility of flanking 50-bp regions of CPAFPs in photosystem-related genes was significantly higher in young leaves and flag leaves than in roots (paired t-test, P value = 1.20e-7 and 1.50e-6, respectively, Fig. 4a, Supplementary Fig. 5a, and Supplementary Table 5), and the FPD scores of CPAFPs for photosystem-related genes were significantly lower in young leaves and flag leaves (paired t-test, P value = 3.8e-8 and 5.8e-13, respectively, Fig. 4a, Supplementary Fig. 5b, and Supplementary Table 6). Consistently, the Tn5 signal of CPAFPs in photosystem-related genes was significantly higher in young leaves and flag

leaves than in root (paired t-test, P value = 2.0e-6 and 2.0e-4, respectively, Fig. 4b). In addition to the photosystem-related genes, the accessibility of flanking 50-bp regions of CPAFPs in rRNA and tRNA genes were also significantly higher in young leaves than in roots (Fig. 4a and Supplementary Fig. 5c).

Chloroplasts are more active in young leaves, so it is to be expected that the genic regions of photosystem-related genes in young leaves exhibit higher accessibility (Fig. 4a). Intriguingly, the FPD scores of CPAFPs in photosystem-related genes in leaves are lower than in roots (Fig. 4b), reflecting a reduced frequency of protein binding. One possible explanation is that proteins bound to genic regions in roots may repress the transcription, as previously reported[19,34], or it is also possible that the transcription is not active enough in roots such that some proteins have more opportunities to bind genic regions and, therefore, generate "deeper" footprints.

To explore the features or potential functions these tissue-differential CPAFPs may have, we first identified 32 CPAFPs with differential FPD scores between roots and young leaves (t-test, P value <0.05, Supplementary Fig. 5d). Then, we extracted the sequencing coverage information of RNA-seq around these CPAFPs. Interestingly, we found that these CPAFPs tended to be distributed in regions where the RNA-seq coverage was altered, suggesting that they may play a role in regulating transcription elongation, termination, or stabilization[21] (Fig. 4c).

**Characterization of CPAFP conservation in grasses**
Since evolutionary conservation reflects functional importance, we analyzed the conservation of CPAFPs identified in the five grasses. We first assessed the proportion of CPAFPs conserved between each two grass species (Methods). CPAFPs were classified into three categories: conserved and identified as footprints in both species, conserved but only identified in one species, and not conserved. Most CPAFPs (~90%) were conserved in sequence between species, and ~40% were conserved in sequence and identified as footprints in both species (Fig. 5a). The large number of conserved CPAFPs indicated that $C_4$ species utilize the regulatory architecture of $C_3$ species not only at the nuclear genome level[35] but also at the chloroplast genome level. We noticed that the proportion of conserved CPAFPs was higher between each two $C_4$ species (S. italica, S. bicolor, and Z. mays) than between $C_4$ and $C_3$ (O. sativa and B. distachyon) (Fig. 5b), indicating that $C_4$ species also shared $C_4$-specific conserved CPAFPs.

By pairwise comparisons within the three $C_4$ species and within the two $C_3$ species, we inspected the proportion of conserved CPAFPs in homologous tissues of different species, and found that the proportion of conserved CPAFPs was remarkably higher in young leaves compared with other tissues in $C_4$ species (Fig. 5b, c). By using sorghum cpDNA as a reference and mapping CPAFPs identified in other species to sorghum cpDNA, we identified 153 conserved CPAFPs that were consistently identified as footprints in young leaves of all three $C_4$ species. Twenty of these CPAFPs could be mapped to the rice genome and overlapped with CPAFPs in young leaves of rice or B. distachyon. Therefore, the remaining 133 CPAFPs were considered to be $C_4$-specific conserved CPAFPs. Based on the gene annotation of sorghum cpDNA, we found that 19.55% (26/133) of the $C_4$-specific CPAFPs were located in or near the photosystem-related genes (Supplementary Fig. 6a), such as genes encoding photosystem II components (psbC and psbD) and cytochrome b/f complex (petB and petD) (Supplemental Data 7, 8), suggesting the photosynthesis of $C_4$ may have evolved more specific and complex regulation. Meanwhile, the genes closest to the conserved footprints

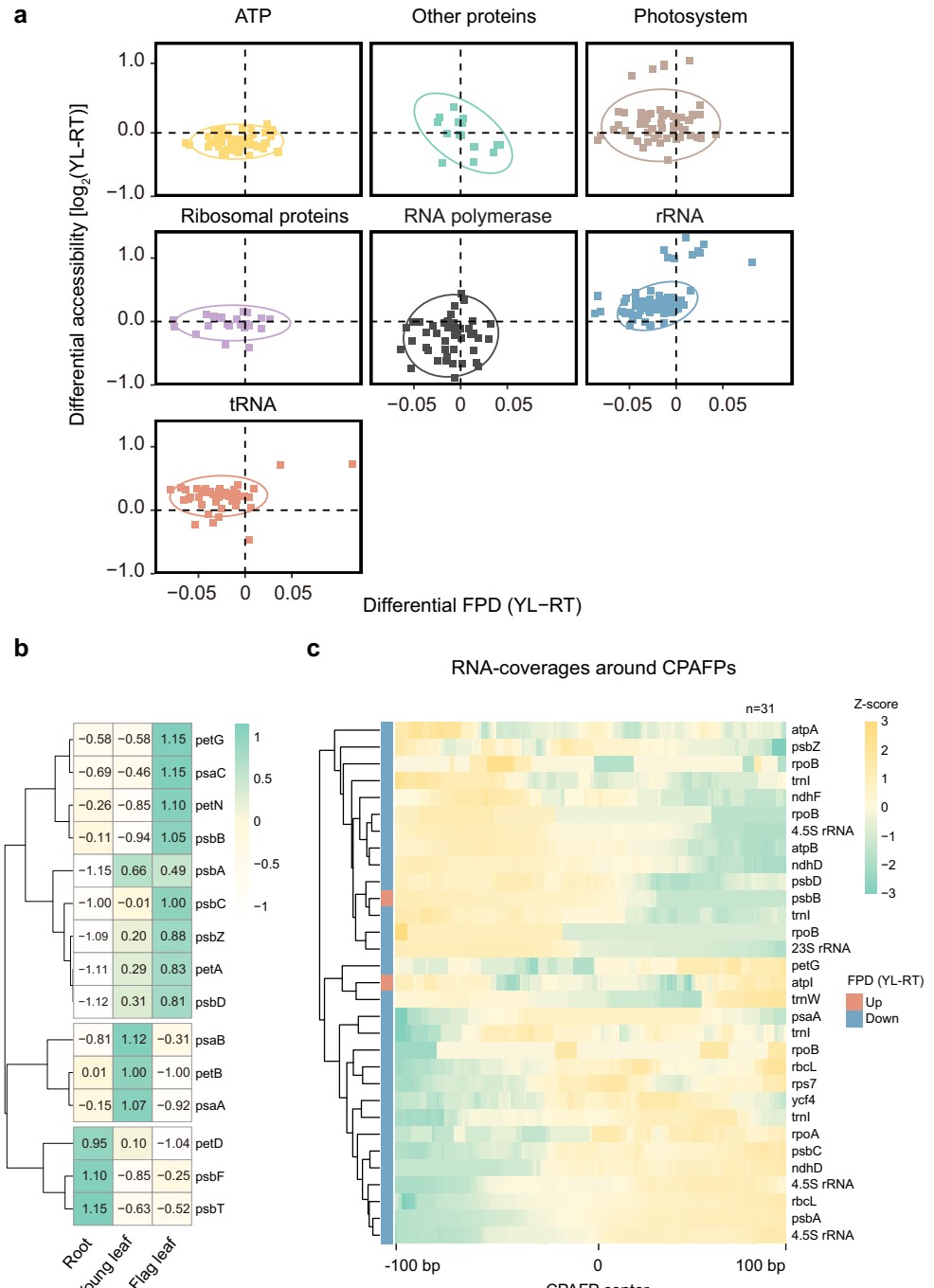

**Fig. 4 | Dynamics of accessibility and CPAFPs between leaves and roots in rice.**
**a** Differences in FPD scores and accessibility of CPAFPs between young leaves and roots on genes with different functional classifications. The x-axis represents differences in FPD scores between young leaves and roots. The y-axis represents differences in accessibility of the flanking 50-bp from CPAFP centers (100-bp in total) between young leaves and roots. The 421 CPAFPs located in genic regions were used here. **b** Average accessibility of the CPAFPs located in the photosystem-related genes in roots and leaves. The accessibility of multiple CPAFPs located in the same photosystem genes was averaged. **c** RNA-seq coverages around tissue-differential CPAFPs. Thirty-two CPAFPs located in genic regions and with differential FPD scores ($t$-test, $P$ value <0.05) between roots and young leaves were selected. Since the RNA-seq coverage of one CPAFP was 0 at each position, 31 CPAFPs were plotted finally.

between $C_3$ and $C_4$ species mainly include rRNA and tRNA genes, *rpoB* and *rpoC* (encoding the RNA polymerase), *psaB* (encoding the core subunit of photosystem I), and genes encoding certain oxidoreductases (*nadhB*, *nadhD*, and *nadhH*). It is noteworthy that research on the regulation of the genes within cpDNA has been relatively sparse. Our findings suggest that investigating the DNA accessibility and transcriptional regulation within cpDNA might be a promising avenue to explore the differences in photosynthesis between $C_3$ and $C_4$ species.

To better explore the features of sequence conservation of CPAFPs, we collected additional cpDNA of three dicots (*Arabidopsis thaliana, Solanum lycopersicum, Solanum tuberosum*) together with the five grasses to obtain the Genomic Evolutionary Rate Profiling (GERP) conserved scores for each site in rice cpDNA. Higher GERP scores indicate greater evolutionary constraints. Therefore, we classified all sites of cpDNA into three classes ("Low", "Middle", and "High") according to the GERP score (Methods). Since the genic and intergenic regions are subject to different evolutionary

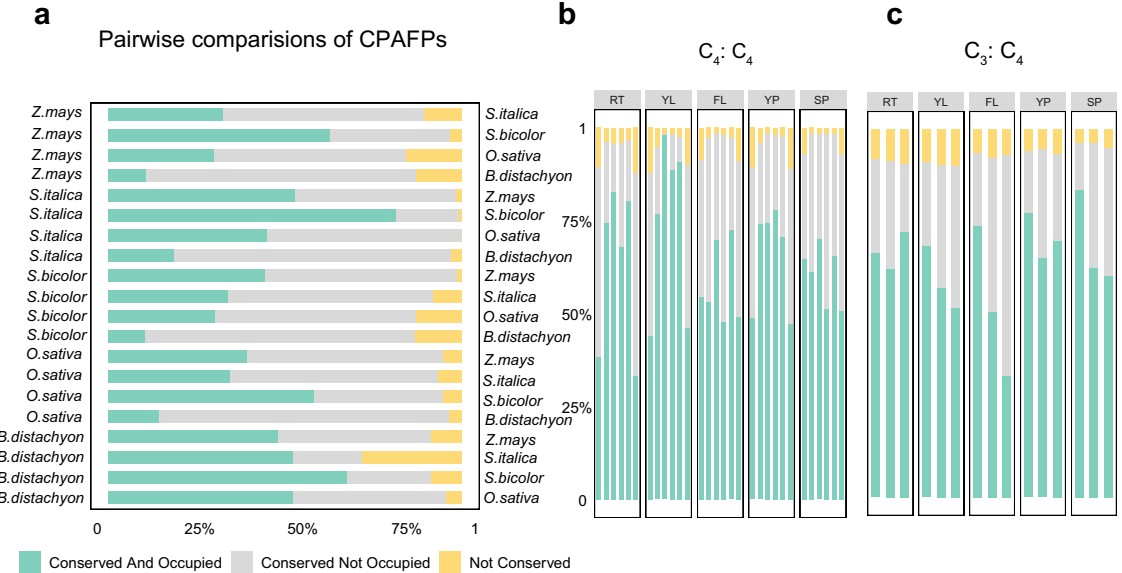

**Fig. 5 | Conservative analysis of conserved CPAFPs in grass species. a** Ratio of conserved footprints between every two species. For each species, CPAFPs identified in different tissues were merged for this analysis. CPAFPs that are conserved and identified as footprints in both species are depicted in green, CPAFPs that are conserved but only identified in one species are depicted in gray shading, and CPAFPs that are not conserved are depicted in yellow. **b** Ratio of conserved footprints between every two $C_4$ species in five homologous tissues. For each species, CPAFPs identified in homologous tissue were used for this analysis. For each tissue, from left to right, the order of comparison is Sb vs. Si, Sb vs. Zm, Si vs. Sb, Si vs. Zm, Zm vs. Sb, and Zm vs. Si. For A vs. B, when calculating the proportion, the number of footprints identified in species A is used as the denominator. RT: Root; YL: Young leaf; FL: Flag leaf; YP: Young panicle; SP: Spikelet. **c** Ratio of conserved footprints between each $C_4$ species and rice ($C_3$ species) in five homologous tissues. For each species, CPAFPs identified in homologous tissue were used for this analysis. For each tissue, from left to right, the order of comparison is Os vs. Sb, Os vs. Si, and Os vs. Zm. Details were the same with (**b**). **d** Distribution of GERP scores at sites covered by rice green tissue-specific CPAFPs versus other sites. CPAFPs identified in young leaves or flag leaves but not in other tissues were defined as green tissue-specific CPAFPs (95 in total). **e** Distribution of GERP scores at sites covered by conserved CPAFPs versus other sites. CPAFPs identified in rice and the homologous positions also identified as footprints in at least one other grass were defined as conserved CPAFPs (407 in total).

constraint, we assessed the distribution of GERP scores at sites covered by different types of CPAFPs in the genic and intergenic regions respectively (Fig. 5d, e, Supplementary Fig. 6b, c, and Supplementary Tables 7, 8). For CPAFPs identified in different tissues in rice, we found that genic sites covered by CPAFPs identified in young leaves have the highest proportion of GERP scores in the "High" class (Supplementary Fig. 6b, c and Supplementary Table 7). This is reasonable because chloroplasts are more active in young leaves, and it is possible that CPAFPs specifically identified in green tissues are more important. To test this possibility, we then identified green tissue-specific CPAFPs by combining the CPAFPs identified in young leaves and flag leaves and excluding those identified in the other tissues and performed the same analysis. We found that the proportions of

"High" class sites covered by rice green tissue-specific CPAFPs were indeed significantly higher than background for both genic and intergenic regions ($P$ value = 5.68e-12 and 9.62e-10, respectively, Fig. 5d), suggesting the biological importance of green tissue-specific CPAFPs. Next, we wondered about the situations in the conserved CPAFPs across species. We found that the proportions of "High" class sites was more significantly increased both at sites covered by CPAFPs identified in rice and at least one other grass (Fig. 5e) or at sites covered by CPAFPs identified in all five grasses (Supplementary Table 8). Taken together, these results further supported that CPAFPs are subject to evolutionary constraints.

## Discussion

In this study, we constructed an accessibility map of cpDNA for five tissues of five grasses and characterized the accessibility patterns and the putative regulatory elements of cpDNAs. Compared to mammalian mtDNAs (~16 kb), cpDNAs of grasses are much larger and more complex (130–140 kb), providing an opportunity to systematically study the relationship between accessibility, footprints, and gene expression in organelles, as well as the characteristics of footprints.

Our findings demonstrate that different regions of cpDNA exhibit significant differences in DNA accessibility, and such differences are related to the structure and expression levels of genes. We identified thousands of putative protein binding footprints on cpDNAs of five grasses and found that these footprints overlap with known functional elements in chloroplasts. In addition, many footprints in cpDNA vary dynamically among tissues, and their sequences are conserved between species. Overall, the accessibility map we constructed provides insight into the DNA state of cpDNA and the footprints we identified add to the knowledge of the functional elements on cpDNA. These resources may accelerate the study of chloroplast regulatory networks.

Intriguingly, we found that the coding region also has a large number of CPAFPs and they tend to overlap non-degenerate sites and contain a higher proportion of highly conserved sites, indicating they are subject to evolutionary constraints. So, how do these CPAFPs function? We found that CPAFPs with differential FPD scores between roots and leaves tend to be distributed in regions where the contextual RNA-seq coverage is altered, suggesting that they may play a role in regulating transcription elongation, termination, or stabilization. In addition, the intergenic regions in cpDNA are short, and some genes overlap with each other. Therefore, these CPAFPs have a dual role of coding and regulation, which may be a result of adaptation to the limited regulatory regions between genes in cpDNA and the need for complex regulation. Furthermore, if the target genes that CPAFPs regulate are not the host genes where CPAFPs are located, but the surrounding genes, then CPAFPs in the CDS regions may also play a role in restricting the order of genes, which may be a reason for the rather conserved gene order in cpDNA. These speculations require further studies.

We also acknowledge the limitations of our study. Some DNA-binding proteins may dissociate during the experimental process, potentially resulting in an incomplete CPAFP landscape. In addition, since the primary isolation target of our ATAC-seq protocol is the nucleus, there may be a bias toward capturing chloroplasts that form plastid–nuclear complexes[36], which might not reflect the state of all chloroplasts. To address this concern, we conducted a preliminary experiment in which we extracted chloroplasts from rice young leaves for ATAC-seq. The results were consistent with those obtained using the conventional ATAC-seq method (Supplementary Fig. 7). Therefore, we encourage researchers to further analyze the available conventional ATAC-seq data in various tissues and species, which would help validate our findings and unveil additional chloroplast transcriptional regulatory mechanisms. However, for studies specifically aimed at chloroplast, isolating pure chloroplast for ATAC-seq analysis will likely result in a more efficient and comprehensive profiling of DNA accessibility in cpDNA.

Moreover, despite we identified footprints from multiple tissues across grasses species, providing resources for studying potential regulatory elements on cpDNA, it is unclear which kind of proteins these elements may be bound by, thus limiting the construction of a regulatory network of the cpDNA. Especially, the conserved footprints are supposed to play fundamental roles in the regulation of cpDNA, and their binding proteins may also be conserved among species. If this information can be elucidated, we believe it will greatly facilitate chloroplast genome engineering.

In conclusion, we have discovered and characterized the accessibility of cpDNA, providing new insights into the transcriptional regulation of chloroplasts. Our study shows that ATAC-seq can be used to study transcriptional regulation for not only the nuclear genome but also cpDNA, providing a new and generally applicable way to study transcriptional regulation in cpDNA. In addition, given that bacterial genomes, like cpDNA, lack nucleosomes, conducting ATAC-seq experiments in bacterial cells may also yield important insights into the mechanisms of bacterial transcriptional regulation[37].

## Methods

### Plant material, ATAC-seq, and RNA-seq experiments

Rice (*Oryza sativa L*, variety ZS97), *Brachypodium distachyon* (variety Bd21), *Setaria italica* (variety Yugu1), *Sorghum bicolor* (variety JDXBR), *Zea mays* (variety B73) were grown under normal agricultural conditions on the experimental farm of Huazhong Agricultural University, Wuhan, China. Five tissues were collected for each grass: roots and young leaves at the seedling stage, young panicles at the branch primordia differentiation stage (or ears at 2–4 mm length for maize), flag leaves and spikelets (stamen and pistil) at the heading stage, with 2–5 biological replicates for each tissue. ATAC-seq were conducted based on our previously established protocol[38]. In brief, fresh samples were cut into small pieces, and the nuclei were separated by chopping with 500 μL chopping buffer (15 mM Tris-HCl pH 7.5, 20 mM NaCl, 80 mM KCl, 0.5 mM spermine, 5 mM 2-ME, 0.2% TritonX-100). After staining with DAPI, each sample was sorted by BD Aria SORP flow cytometry, and 100,000 nuclei were collected into a 1.5 mL tube. The nuclei were centrifuged and enriched by removing the supernatant, then Tn5 tagmentation was conducted at 37 ℃ for 30 min.

The ATAC-seq data in rice (variety ZS97) has been published in our previous work[39] while data for other species are newly generated. RNA-seq libraries were constructed and sequenced by Novogene, Inc.

### ATAC-seq data processing

Raw reads were first trimmed by Trimmomatic (v.0.36)[40], with parameters of a maximum of two seed mismatches, a palindrome clip threshold of 30, and a simple clip threshold of 10, reads shorter than 30-bp were discarded. Then reads were aligned to the corresponding cpDNA of *Oryza Sativa* (RefSeq: NC_001320.1), *Brachypodium distachyon* (RefSeq: NC_011032.1), *Setaria italica* (RefSeq: NC_022850.1), *Sorghum bicolor* (RefSeq: NC_008602.1), *Zea mays* (RefSeq: NC_001666.2) using bwa mem algorithm[41]. Mapped reads with a MAPQ quality score below 0 and PCR duplicates were filtered using SAMTools (v.1.9)[42] to ensure high-quality aligned data. The Tn5 insertion positions were determined as the start sites of reads and adjusted by the rule of "forward strand +4 bp, negative strand –5 bp". The cpDNA annotation of five species were also obtained from the RefSeq id mentioned above.

To quantify the accessibility, we divided the cpDNA into 100-bp bins. For each ATAC-seq dataset, the accessibility was quantified by counting Tn5 insertions per bin. Correlations between ATAC-seq replicates were calculated based on accessibility quantification results (Supplementary Fig. 1d).

### RNA-seq data processing

Raw reads in FASTQ format were mapped to the cpDNA using STAR 2.7.7a with two-pass mode[43]. The coverage of the bam files was extracted using the bamCoverage command with 100-bp bins and normalized. Based on the cpDNA annotations, we calculated expression levels (TPM) of the

annotated genes using Salmon[44]. Differentially expressed genes between tissues were identified by DESeq2[45].

## CPAFPs identification

The bam files of samples with replicates were merged using "samtools merge". Then CPAFPs for each sample were identified using "wellington_footprints.py" in pyDNase[25] with default parameters (-fdr 0.01) except for "-A" mode. In all analysis, CPAFPs that overlap 1 bp were defined as shared CPAFPs and were obtained using "bedtools intersect" with default parameters[46].

## Evaluating the effect of Tn5 bias on footprinting

We performed Tn5 tagmentation on purified genomic DNA (gDNA) of rice young leaves, obtained a similar number of reads as ATAC-seq data of rice young leaves (1,967,428 for ATAC and 1,801,447 for gDNA) and utilized gDNA data to evaluate the effect of Tn5 bias on the identification of footprints.

$$\text{Corrected}_i = \frac{\text{Raw}_i}{\text{gDNA}_i + 1}$$

The number of Tn5 insertions at each base was corrected using the above equation, where "i" is a position of the chloroplast genome, "Raw" represents the number of insertions in the original ATAC data, and "Correct" represents the corrected number of insertions. Sites with similar Tn5 insertions in ATAC and gDNA should have signals close to 1 after the correction. After obtaining the corrected Tn5 signal at each position, for every footprint identified by ATAC-Seq data, the footprint depth scores (FPDs, refers to Fig. 3a) were calculated using the following equation from both the raw and corrected data separately. Where FKI is the total number of Tn5 insertions in the flanking regions (50 bp on each side of the footprint); FPI is the total number of Tn5 insertions within the footprint, and L is the length of the footprint. A larger FPD score could be interpreted as a reduction in Tn5 insertions in the footprint region compared to the flanking regions.

$$\text{FPD} = \left(\frac{\text{FKI}}{100} - \frac{\text{FPI}}{L}\right) / \left(\frac{\text{FKI}}{100} + \frac{\text{FPI}}{L}\right)$$

## Analysis of contamination of nuclear plastid DNAs (NUPTs)

To confirm the impact of NUPTs in estimating accessibility and identifying footprints, we calculated the proportion of NUPTs in each sample. Based on the variations between cpDNA and sequences transferred into the nucleus, we developed a method similar to allele-specific expression (ASE) analysis to calculate the proportion of NUPTs by referring to similar studies. Briefly, we first ran GATK to call SNPs in the sequences that could align to both cpDNA and the nuclear genome. Then we calculated the coverage of "ref allele" and "alt allele" for each SNP (by GATK ASEReadCounter)[47], which represent the sequences from cpDNA and the nuclear genome, respectively. The NUPT ratio was calculated as the sum of the coverage of the "alt allele" at all sites divided by the sum of "all alleles" at all sites.

## Conservative analysis of footprints

To identify conserved footprints across multiple species, we first merged the footprints shared by different tissues in each species using the command "bedtools merge", i.e., footprints that overlap 1 bp will be merged. For the two species, A and B, to be compared (B as the reference species), we extracted the 50-bp sequence in the cpDNA of A centered on the identified footprints and performed BLAST with the cpDNA of B. Footprints with BLAST score >85 and alignment length >45-bp were considered as conserved footprints between A and B. To obtain conservation scores for each site in cpDNA, in addition to the cpDNAs of the five grasses, we also collected cpDNAs of three dicots (NC_000932.1 for

*Arabidopsis thaliana*, NC_007898.3 for *Solanum lycopersicum* and NC_008096.2 for *Solanum tuberosum*) for calculating conservative scores. We first performed multiple sequence alignment for the eight cpDNAs and obtained a phylogenetic tree using MAFFT[48]. Next, GERP conservation scores for each site in cpDNA were calculated using the 'PhyloP' program in the PHAST package[49]. Finally, GERP scores were categorized into three classes as "Low" (GERP score <0, accounted for 24.6%), "Middle" (0 < = GERP score <0.42, accounted for 21.7%), and "High" (GERP score = 0.42, since the max score is 0.42, accounted for 53.7%).

## Fourfold degenerate site (4DS) analysis

A codon is composed of three nucleotides. For the third site of a codon, if all substitutions are synonymous, this site is called a 4DS. 4DSs are generally considered to be less constrained by selection. We identified all potential 4DSs and non-4DSs at the third position of codons in CDS regions of cpDNA with a PERL script (https://github.com/tsackton/linked-selection/blob/master/misc_scripts/Identify_4D_Sites.pl)[50] according to the chloroplast codon table and annotations. Then, 4DSs in the regions of interest (such as CPAFPs) were counted respectively. Finally, the counted 4DS number were subjected to do Fisher's exact test to evaluate the distribution preference.

## Statistics and reproducibility

All statistical analyses were conducted using the packages listed in the Methods section. The *T*-tests were performed in R using the t.test function, while correlations between datasets were calculated using Pearson correlation coefficients. The CPAFPs distribution preference was calculated by Fisher's exact test by R. The conservative analysis of conserved CPAFPs across grasses was conducted by Z-test in R. The data used in this study is available on NCBI, and the results can be reproduced based on the parameters outlined in the Methods section.

## Reporting summary

Further information on research design is available in the Nature Portfolio Reporting Summary linked to this article.

## Data availability

Sequencing data from this study will be available in the NCBI sequence read archive (SRA) database under the BioProject accession number PRJNA705005. The source data behind the graphs in the paper can be found in Supplemental Data 1–8.

## Code availability

Codes for any necessary to reproduce the analyses are available upon request.

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

## Acknowledgements

The computations in this paper were run on the bioinformatics computing platform of the National Key Laboratory of Crop Genetic Improvement, Huazhong Agricultural University. This work was supported by grants from the National Key R&D Program of China (2023ZD04073 and 2023ZD04076), the National Natural Science Foundation of China (U23A20189, 32261143466, and 31771755), the Hubei Provincial Natural Science Foundation of China (2023AFA043), the Fundamental Research Funds for the Central Universities (2662023PY002), and the Earmarked Fund for the China Agriculture Research System (CARS-01-01).

## Author contributions

W.X. designed and supervised this study. Jiacheng. L. and F.Z. provided ideas for data analysis. Y.L. and Junji. L. performed the bioinformatics analysis. Y.L. and J.Z. wrote the manuscript. W.X. and M.L. revised the manuscript. C.X. conducted the experiments. All the authors read and approved the manuscript.

## Competing interests

The authors declare no competing interests.
