## [Peer Review File · Communications Biology]

Reviewers' comments:

Reviewer #1 (Remarks to the Author):

This study used the ATAC-seq technique to explore the regulation of cpDNA. Although this may be the first time of ATAC-seq was used in grasses and they have some interesting findings, I can't find a specific scientific question to be proposed in the introduction and revolved in methods and discussion. I have to admit that the workflow for the application of ATAC-seq in cpDNA, but this is research just focused on a technology development that had been used in mtDNA. Although the application range of this technique would be extended from this research, I think this paper is suitable for a more specific journal about the development of technology.

Here are some minor comments.

Line 62: Does the accessibility mean "chromatin accessibility"? if yes or no, it is necessary to give a full name and explain what accessibility is. It is tough to understand the meaning of a single "accessibility".

Line 148: downstream is downstream?

Line 426 "...cpDNA is short" should "...cpDNA are short"

Line 438: "fundmental" is "fundamental"?

Line 457: there is no reference "Zhao et al. 2021" in the reference list

Reviewer #2 (Remarks to the Author):

In this study, by generating ATAC-seq data for multi-tissues of five grasses, the authors constructed a high-resolution accessibility map of cpDNA and characterized the accessibility of cpDNA of five grasses for the first time. This study shows that ATAC-seq can be used to study transcriptional regulation for not only the nuclear genome but also cpDNA, providing a new and generally applicable way to study transcriptional regulation in cpDNA. I suggest accepting this manuscript.

But there are still several minor questions to be answered :

1. Has the ATAC-seq data in rice been published in other journals? Need to explain
2. Please describe in more detail the differences in chloroplast genome expression between C3 plants and C4 plants

Reviewer #3 (Remarks to the Author):

The authors provide for the first time data about the transposase accessibility of chloroplast genomes. This nicely shows that there are differences in accessibility between different genomic regions and that these patterns are flexible between different species and tissues suggesting the reflection of altered DNA-based regulation of gene expression. In comparison to RNA seq data, the ATAC-seq data show that cpDNA accessibility correlates with expression level. Their data provide nice insights with high potential links to biological functions. I have two major concerns and some minor points that should be addressed before publication:

Major points:

1. Please describe better in the Methods how the data were achieved. You just refer to your previous publication in which you applied ATAC-seq on purified nuclei. How can that method provide chloroplast data? Is there a high degree of chloroplast contamination in your nuclei-enriched samples (and is this contamination reproducible between samples)? Or is it possible that the chloroplast sequences, at least in part, derive from nuclear copies of chloroplast DNA (NUPTS) (you can probably rule that out by the missing nucleosome pattern – but it is not mentioned)? If the sequences derive from genuine chloroplast DNA, how does the transposase enter the chloroplast? Or do the data derive from cp-DNA that was released from broken chloroplasts and co-purified with nuclei? If the latter is the case, which fraction of the cpDNA-binding proteins would still be bound? It would likely require some additional test experiments to disentangle these possibilities and to make sure from which source your data are. At least, provide more details discussing these questions in the Methods (and potentially Results) sections.
2. Figure 2a shows higher accessibility in intergenic regions than in genic regions (same for footprint abundance shown in Table 1). Chloroplast genes are organized in operon-like arrays of genes that are co-transcribed and post-transcriptionally processed. In terms of transcription, the intergenic regions between these co-transcribed genes are rather “genic” than “intergenic”. Did you account for that in your analysis? The situation is comparable to that in bacteria (due to the endosymbiotic origin of chloroplasts) – how are ATAC-seq data interpreted in bacteria (in this perspective)?

Minor points:

1. Line 1 (title): It may be worth to think about a title that more precisely describes the results. “[...] accessibility of chloroplast genomes” could have many different meanings...
2. Line 30 (key words): “accessibility” may be not very informative as key word. Maybe “DNA accessibility”?
3. line 51: explain in more detail what you mean by “DNA state of cpDNA” (e.g., structure, confirmation, etc.).
4. line 59: “we” should be replaced by “one” or the sentence should be rephrased to make clear that this method was not invented by you.
5. line 66: “multi-tissues” should be “multiple tissues”.
6. line 122: “high” should be “highly”
7. line 150: citations 19-20; there is at least one alternative interpretation available for the action of a specific cpmTERF protein (Meteignier, L.V., et al., The Arabidopsis mTERF-repeat MDA1 protein plays a dual function in transcription and stabilization of specific chloroplast transcripts within the psbE and ndhH operons. *New Phytol*, 2020. 227(5): p. 1376-1391.)
8. line 176: In the first sentence the information is missing that this was applied in different species.
9. lines 238-240:/Figure 2e: I see the described reduced accessibility only at -10 but not at -35. Any explanation for that?
10. lines 246-250: could you calculate a number for accessibility and do statistics on that to demonstrate that in the ori region of the different species the accessibility is indeed statistically significantly reduced?
11. line 289: “reglations” should be “regulation”

12. line 317-320 and 424-426: Due to the low turnover of cp transcripts, most of the transcripts detected in RNA seq are not reflecting the transcriptional status but rather the post-transcriptional RNA processing status – how that is related with DNA-binding proteins may be potentially interesting (one example for a potential connection between DNA-binding and RNA processing may be given in Meteignier, L.V. et al. 2020). I disagree with the pure statement made here that this could reflect regulation in transcription elongation/termination without mentioning RNA processing status (which is what you measure in cpRNA-seq).

13. line 436/437: “thus limiting the construction of a regulatory network of the cpDNA”.

14. line 438/439: “may also be conserved among species”.

Reviewer #4 (Remarks to the Author):

In this study presented by Liu et al., the authors compared chloroplast ATAC-seq reads generated from a group of diverse tissues and different grass species. A positive association between highly accessible regions and active chloroplast gene expression is confirmed. Furthermore, an association between footprints that imply protein-cpDNA interactions and known transcriptional regulatory elements is presented in this manuscript.

However, there is a potential technical issue, which in my opinion, should be addressed before analyzing these datasets in depth. According to the “Method” section, the chloroplast ATAC-seq reads were recovered from a regular ATAC-seq protocol, in which nuclei were isolated with a FACS machine. It is a clever idea to use chloroplast-derived reads from such an ATAC-seq experiment and to infer cpDNA “accessibility”. Nonetheless, it is critical to understand the nature of cpDNA co-isolated with nuclei. Regarding this point, I have the following two concerns:

- 1) Were these chloroplasts intact or damaged (or fragmented) at the time point of starting Tn5 reactions? If they were damaged, to which extent could the cpDNA accessibility be potentially affected, for example, as a result of interactions between cpDNA and other cellular components present in the homogenate?
- 2) The experimental procedure was designed to harvest nuclei. Is it true that the cpDNA can be treated as a fair presentation of the entire chloroplast population of the tissue?

Reviewer #1 (Remarks to the Author):

This study used the ATAC-seq technique to explore the regulation of cpDNA. Although this may be the first time of ATAC-seq was used in grasses and they have some interesting findings, I can't find a specific scientific question to be proposed in the introduction and revolved in methods and discussion. I have to admit that the workflow for the application of ATAC-seq in cpDNA, but this is research just focused on a technology development that had been used in mtDNA. Although the application range of this technique would be extended from this research, I think this paper is suitable for a more specific journal about the development of technology.

Response: We appreciate the reviewer's recognition of our novel findings and fully understand the reviewer's concern.

Firstly, we would like to clarify that our study does have specific scientific questions, which are to investigate the accessibility patterns of cpDNA in grasses and to understand its biological implications for transcriptional regulation in chloroplasts. This question is driven by the limited knowledge on cpDNA state and regulatory elements, as mentioned in the introduction (lines 39-54).

Secondly, as we mentioned in the introduction, Blumberg et al. first characterized the accessibility and footprinting patterns of mtDNAs in mammals using DNase-seq data. Inspired by their interesting findings, we extend the application of this approach to cpDNAs, which are much larger and more complex (130-140 kb) in grasses compared to mammalian mtDNAs (~16 kb). This expansion allowed us to uncover novel insights into chloroplast transcriptional regulation, such as:

- (1) The significant differences in accessibility of different cpDNA regions and their biological implications.
- (2) The identification of 3,970 putative protein binding footprints on cpDNA in grasses, enriched in intergenic regions and co-localized with known functional elements.
- (3) The presence of footprints in coding regions that overlap non-degenerate sites and contain highly conserved sites, indicating evolutionary constraints.
- (4) The dynamic variation of footprints and their flanking accessibility in different tissues, associated with changes in expression.

Therefore, our study includes not only the technological aspect but also provides valuable biological findings, which are fitting for the subject matter of *Communications Biology*.

In light of the reviewers' comments, we have revised the manuscript to more clearly emphasize the scientific question and the biological implications of our findings in the introduction, and discussion sections (Lines 51-78 and 426-454).

Here are some minor comments.

Line 62: Does the accessibility mean "chromatin accessibility"? if yes or no, it is

necessary to give a full name and explain what accessibility is. It is tough to understand the meaning of a single “accessibility”.

Response: We thank the reviewer for pointing out this problem in the manuscript. As Reviewer#3 suggested, we have replaced it to “*DNA accessibility*” in the revised manuscript. This term is not only more explicit but also commonly used in many studies.

Line 148: downstream is downstream?

Response: We have corrected it in Line 152 as “... *at the downstream of GESs*”.

Line 426 “...cpDNA is short” should “...cpDNA are short”

Response: We have corrected it in Line 447 as “*the intergenic regions in cpDNA are short*”.

Line 438: “fundmental” is “fundamental”?

Response: We have corrected it in Line 464 as “... *play fundamental roles in ...*”.

Line 457: there is no reference “Zhao et al. 2021” in the reference list.

Response: Sorry for this mistake, it has now been added as reference [39] in Methods section Lines 491-493.

Reference:

Zhao H, et al. An inferred functional impact map of genetic variants in rice. *Molecular Plant*, 2021, 14: 1584-1599.

Reviewer #2 (Remarks to the Author):

In this study, by generating ATAC-seq data for multi-tissues of five grasses, the authors constructed a high-resolution accessibility map of cpDNA and characterized the accessibility of cpDNA of five grasses for the first time. This study shows that ATAC-seq can be used to study transcriptional regulation for not only the nuclear genome but also cpDNA, providing a new and generally applicable way to study transcriptional regulation in cpDNA. I suggest accepting this manuscript.

Response: We thank the reviewer for the positive comments on our study.

But there are still several minor questions to be answered :

1. Has the ATAC-seq data in rice been published in other journals? Need to explain

Response: The ATAC-seq data in rice (ZS97 variety) has been published in 2021 as reference [39] and the rest data have not been published yet. We have updated detailed description of our data in Methods section in Lines 491-493 as “*The ATAC-seq data in rice (variety ZS97) has been published in our previous work while data for other species are newly generated*”.

Reference:

Zhao H, et al. An inferred functional impact map of genetic variants in rice. *Molecular Plant* 2021, 14: 1584-1599.

2. Please describe in more detail the differences in chloroplast genome expression between C3 plants and C4 plants.

Response: We thank the reviewer for the suggestion. Given that our study aimed to characterize the DNA accessibility within cpDNA rather than gene expression, and considering the lack of relevant data, we have incorporated additional descriptions in the Results section (Lines 366-378) and included Supplemental Data to detail the conserved footprints in C₃ and C₄ species respectively. Our analyses showed that certain regulatory elements are particularly conserved in C₄ species compared to C₃ species (Fig. 5a-b). We found that the genes closest to the conserved footprints between C₃ and C₄ species mainly include rRNA and tRNA genes, *rpoB* and *rpoC* (encoding the RNA polymerase), *psaB* (encoding the core subunit of photosystem I), and genes encoding certain oxidoreductases (*nadhB*, *nadhD*, and *nadhH*). In contrast, the footprints that specific conserved in C₄ plants are mainly close to genes encoding photosystem II components (*psbC* and *psbD*) and cytochrome b/f complex (*pet B* and *petD*) (Supplemental Data). It is noteworthy that research on the regulation of the genes within cpDNA has been relatively sparse. Our findings suggest that investigating the DNA accessibility and transcriptional regulation within cpDNA might be a promising avenue to explore the differences in photosynthesis between C₃ and C₄ species.

Reviewer #3 (Remarks to the Author):

The authors provide for the first time data about the transposase accessibility of chloroplast genomes. This nicely shows that there are differences in accessibility between different genomic regions and that these patterns are flexible between different species and tissues suggesting the reflection of altered DNA-based regulation of gene expression. In comparison to RNA-seq data, the ATAC-seq data show that cpDNA accessibility correlates with expression level. Their data provide nice insights with high

potential links to biological functions. I have two major concerns and some minor points that should be addressed before publication:

Response: We thank the reviewer for the positive comments on our study.

Major points:

1. Please describe better in the Methods how the data were achieved. You just refer to your previous publication in which you applied ATAC-seq on purified nuclei. How can that method provide chloroplast data?

Response: Sorry for the confusion and we have added more detailed description in the Methods (Lines 480-493). Briefly, the ATAC-seq experimental procedure was to first chop the sample with 500 μ L chopping buffer to release the nuclei. After staining with DAPI, 100,000 nuclei were sorted into 1.5 ml tubes using flow cytometry. Nuclei were centrifuged and enriched by removing the supernatant, followed by Tn5 tagmentation at 37°C. Since the chopping buffer contains 0.2% detergent Triton X-100, pores may be formed in the nuclei and chloroplast membranes that facilitate the entry of Tn5.

Although this procedure is designed to reduce the organelle content and has been applied to obtain nuclei for ATAC-seq experiments in both plants and animals, studies have shown that the organelle content cannot be completely removed through the procedure, the mapping rate of sequencing reads to organelle are 0.86%-5.46% for *Zebrafish*, 2.5%-20% for *Arabidopsis thaliana* respectively in published studies (Lu et al., 2017; Quillien et al., 2017). Therefore, due to the high copy number of cpDNA in cells and its small size, we were still able to obtain high coverage of chloroplast data.

Reference:

Lu Z, et al. Combining ATAC-seq with nuclei sorting for discovery of cis-regulatory regions in plant genomes. *Nucleic Acids Research*, 2017, 45: e41-e41.

Quillien A, et al. Robust Identification of Developmentally Active Endothelial Enhancers in Zebrafish Using FANS-Assisted ATAC-Seq. *Cell Reports*, 2017, 20: 709-720.

Is there a high degree of chloroplast contamination in your nuclei-enriched samples (and is this contamination reproducible between samples)? Or is it possible that the chloroplast sequences, at least in part, derive from nuclear copies of chloroplast DNA (NUPTS) (you can probably rule that out by the missing nucleosome pattern – but it is not mentioned)?

Response: According to our statistical results, the proportion of reads mapped to cpDNA varies from 0.1% to 29% (with an average coverage of 1000X for cpDNA)

across different tissues and species, and the details could be found in Supplemental Table 1. The cpDNA sequence ratios were found to be relatively consistent across biological replicates in all species but varied between tissues. For example, in *S. italica*, the cpDNA mapping rates were 11.17% and 12.65% for two replicates of flag leaf tissue, respectively, while for panicle tissue, the rates were 4.02% and 5.77% for two replicates, respectively.

Regarding the potential contribution of NUPTs, our analysis indicates that some mapped sequences are indeed derived from NUPT. By calculating the proportion of NUPT reads across all samples, we confirmed that the proportion is relatively small and has negligible impact on our analysis (Please refer to Results section “Identifying accessibility footprints of chloroplast genomes in five grasses” in Lines 210-221 and Method section “Analysis of contamination of nuclear plastid DNAs” in Lines 539-549 for detail).

In response to the suggestion to examine the nucleosome pattern, we plotted the fragment distribution of reads mapped to chloroplasts and the results clearly indicated the absence of nucleosome pattern (pictures are attached below), supporting that these sequences are derived from cpDNA (Supplementary Fig. 1).

If the sequences derive from genuine chloroplast DNA, how does the transposase enter the chloroplast? Or do the data derive from cp-DNA that was released from broken chloroplasts and co-purified with nuclei? If the latter is the case, which fraction of the cpDNA-binding proteins would still be bound? It would likely require some additional test experiments to disentangle these possibilities and to make sure from which source your data are. At least, provide more details discussing these questions in the Methods (and potentially Results) sections.

Response: Thank you for your insightful comments. Briefly, the experimental procedure was to first chop the sample with 500 μ L chopping buffer to release the nuclei. After staining with DAPI, 100,000 nuclei were sorted into 1.5 ml tubes using

flow cytometry. Nuclei were centrifuged and enriched by removing the supernatant, followed by Tn5 tagmentation at 37°C. Since the chopping buffer contains 0.2% detergent Triton X-100, pores may be formed in the nuclei and chloroplast membranes that facilitate the entry of Tn5. Some chloroplasts may be co-purified with nuclei, as plant cells have plastid–nuclear complexes (Selga et al., 2010). Although this process removes most of the broken chloroplasts, our data likely contains a mixture of both intact and broken chloroplasts. However, even chloroplasts are broken, some proteins may continue to bind to cpDNA. As many recent techniques, such as native ChIP-seq (Kasinathan et al., 2014) and CUT&Tag (Kaya-Okur et al., 2019), do not require cross-linking when studying DNA-protein interactions, suggesting that many proteins can remain bound to DNA under uncross-linked conditions. In our study, the identified footprints were co-localized with known functional elements, providing support for our results. Furthermore, similar experimental conditions have been used to study mtDNA (Blumberg et al., 2018), suggesting that our approach is feasible. However, we acknowledge that some proteins may have dissociated from cpDNA during this process and their binding signals may not be detected. We have added more detailed description in the methods section (Lines 480-493) and a relevant discussion in Lines 455-466.

Reference:

Blumberg A, et al. A common pattern of DNase I footprinting throughout the human mtDNA unveils clues for a chromatin-like organization. *Genome Research*, 2018, 28: 1158-1168.

Kasinathan S, et al. High-resolution mapping of transcription factor binding sites on native chromatin. *Nature Methods*, 2014, 11: 203-209.

Kaya-Okur HS, et al. CUT&Tag for efficient epigenomic profiling of small samples and single cells. *Nature Communications*, 2019, 10: 1930.

Selga T, et al. Plastid-nuclear complexes: permanent structures in photosynthesizing of vascular plants. *Environmental and Experimental Biology*, 2010, 8: 85-92.

2. Figure 2a shows higher accessibility in intergenic regions than in genic regions (same for footprint abundance shown in Table 1). Chloroplast genes are organized in operon-like arrays of genes that are co-transcribed and post-transcriptionally processed. In terms of transcription, the intergenic regions between these co-transcribed genes are rather “genic” than “intergenic”. Did you account for that in your analysis? The situation is comparable to that in bacteria (due to the endosymbiotic origin of chloroplasts) – how are ATAC-seq data interpreted in bacteria (in this perspective)?

Response: In our analysis, we have considered the operon structure within the chloroplast genome. As stated in Lines 151-159 of the revised manuscript (Lines 147-152 of the original manuscript), we did not consider the structure of the operons in cpDNA in our initial analysis (Figure 2a), as studies have shown that most genes within the operons also have promoters (Zhelyazkova et al., 2012). After observing the higher accessibility in intergenic regions, to further confirm the results, we collected 43 reported operons (Kanno A et al., 1993; Zhelyazkova et al., 2012; Shi et al., 2016) and redefined genic and intergenic regions based on the start and end sites of the operons, and observed similar results (Supplementary Fig. 2). The results in Table 1 have considered the structure of the operons.

In response to the reviewer's suggestion, we conducted the same analysis as in Figure 2a by redefining the genic and intergenic regions based on the 43 reported operons and added Supplementary Fig. 2a (below). The results show that the intergenic regions remain less accessible than the genic regions, as shown in Figure 2a.

Supplementary Fig. 2a. The distributions of accessibility in genic (43 reported operons) and intergenic regions. The x-axis is the average Tn5 insertions of per region (first sum the Tn5 insertions for each genic and intergenic regions, then divided by its length), while y-axis represents the density.

Regarding the second question, performing similar experiments in bacteria should also contribute to the understanding of bacterial transcriptional regulation. In fact, a study has already been conducted, and according to their results, the *E. coli* genome also

exhibited the DNA accessibility differences in different regions (Mahmoud et al., 2020). We have added relevant discussions in Lines 471-474.

Reference:

Kanno A, et al. A transcription map of the chloroplast genome from rice (*Oryza sativa*). *Current Genetics*, 1993, 23: 166-174.

Mahmoud MA-B, et al. Nucleoid openness profiling links bacterial genome structure to phenotype. *bioRxiv*, 2020, 082990.

Shi C, et al. Full transcription of the chloroplast genome in photosynthetic eukaryotes. *Scientific Reports*, 2016, 6: 30135.

Zhelyazkova P, et al. The Primary Transcriptome of Barley Chloroplasts: Numerous Noncoding RNAs and the Dominating Role of the Plastid-Encoded RNA Polymerase. *The Plant Cell*, 2012, 24: 123-136.

Minor points:

1. Line 1 (title): It may be worth to think about a title that more precisely describes the results. “[...] accessibility of chloroplast genomes” could have many different meanings...

Response: Thanks for your advice, it has been updated as “*DNA accessibility of chloroplast genomes*” now.

2. Line 30 (key words): “accessibility” may be not very informative as key word. Maybe “DNA accessibility”?

Response: Thanks for your advice, it has been updated as “*DNA accessibility*” now.

3. line 51: explain in more detail what you mean by “DNA state of cpDNA” (e.g., structure, confirmation, etc.)

Response: Thanks for your suggestion, it has been updated as “*the DNA state of cpDNA (e.g., binding state of transcription factors or other proteins and the DNA accessibility)*” now.

4. line 59: “we” should be replaced by “one” or the sentence should be rephrased to make clear that this method was not invented by you.

Response: Thanks for your valuable suggestion, it has been corrected now in Line 61 as “*By comparing the distribution of Tn5 insertions, one can identify such footprints, which indicate potential regulatory elements*”.

5. line 66: “multi-tissues” should be “multiple tissues”.

Response: Thanks for your valuable suggestion, it has been corrected now in Line 70 as “*by generating ATAC-seq data for multiple tissues of five grasses*”.

6. line 122: “high” should be “highly”

Response: Thanks for your valuable suggestion, it has been corrected now in Line 127 as “*We clearly observed that highly accessible regions of cpDNA*”.

7. line 150: citations 19-20; there is at least one alternative interpretation available for the action of a specific cpmTERF protein (**Méteignier, L.V., et al., The Arabidopsis mTERF-repeat MDA1 protein plays a dual function in transcription and stabilization of specific chloroplast transcripts within the psbE and ndhH operons. New Phytol, 2020. 227(5): p. 1376-1391.**)

Response: Thanks for your valuable suggestion, we have checked the literature carefully and added it in the revised manuscript as reference [21] in Lines 328-331 to support this idea.

Reference:

Méteignier L-V, et al. The Arabidopsis mTERF-repeat MDA1 protein plays a dual function in transcription and stabilization of specific chloroplast transcripts within the psbE and ndhH operons. New Phytologist, 2020, 227: 1376-1391.

8. line 176: In the first sentence the information is missing that this was applied in different species.

Response: Thanks for your valuable suggestion, we have updated it as “*We further conducted footprinting analysis to screen the potential regulatory elements in cpDNA across all five grasses*” in Lines 182-183.

9. lines 238-240:/Figure 2e: I see the described reduced accessibility only at -10 but not at -35. Any explanation for that?

Response: As demonstrated in Figure 3e, the decline in accessibility is not as pronounced in the -35 region as in the -10 region. A potential explanation aligns with

the observations made by Zhelyazkova et al (2012) in their barley study. They identified typical motifs of PEP promoters upstream of transcription start sites, with the -10 and -35 elements being mapped at rates of 89% and 70%, respectively. They proposed that the lower conservation rate of the -35 elements suggested that PEP transcription may proceed without the -35 element, possibly relying on motifs that are substantially dissimilar to the *E.coli* σ^{70} -35 element or other variable cis-elements. This suggests that the -10 element, being more conserved, is crucial for chloroplast transcription initiation and might be subject to more frequent protein binding. We have clarified this point in the Results section in Lines 247-250 as “*A possible explanation is that the -10 element is more conserved compared to -35 element and plays a more important role in the transcription of chloroplast genes, therefore it might be bound more frequently*”.

Reference:

Zhelyazkova P, et al. The Primary Transcriptome of Barley Chloroplasts: Numerous Noncoding RNAs and the Dominating Role of the Plastid-Encoded RNA Polymerase. *The Plant Cell*, 2012, 24: 123-136.

10. lines 246-250: could you calculate a number for accessibility and do statistics on that to demonstrate that in the ori region of the different species the accessibility is indeed statistically significantly reduced?

Response: In this study we used pyDNase (Piper J., et al 2013) to identify footprints. As the reviewer suggested, we calculated the DNA accessibility of footprints in ori regions across different species as follows (ATAC-seq data of young leaf was used). The results show that within the ori region, the accessibility of identified footprints is significantly reduced compared to the flanking regions.

The Wellington *P*-value is given by pyDNase (Piper J., et al 2013): “With $F(k, n, p)$ being the binomial cumulative distribution function, i.e. the probability of achieving at least k out of n successes with the probability of each success being p , we calculate a *p*-value using the formula:

$$P - value = F\left(FP^+, FP^+ + SH^+, \frac{l_{FP}}{l_{FP} + l_{SH}}\right) \times F\left(FP^-, FP^- - SH^-, \frac{l_{FP}}{l_{FP} + l_{SH}}\right)$$

This *p*-value is for a given possible footprint of length l_{FP} with surrounding shoulder regions of length l_{SH} .” l_{FP} is the length of a possible footprint; l_{SH} is the length of shoulder on each side of the footprint; FP^+ (forward reference strand) and FP^- (negative reference strand) are the total Tn5 insertions inside the footprint. SH^+ (upstream

shoulder on the forward reference strand) and SH⁻ (downstream shoulder on the negative strand) are the total Tn5 insertions inside the shoulder of the footprint. F is the binomial cumulative distribution function. Wellington has a command line argument to employ an empirical method of estimating the False Discovery Rate (FDR). Briefly, it shuffle the number of tags aligned to each base pair within a hyper-sensitive site (in our research it refers to the whole chloroplast genome) and recalculate the “Wellington *P*-value” on this shuffled data 500 times, and then can determine a p-value threshold which would only occur at most 1 in 100 times, corresponding to an FDR of 0.01.

Species	Start (bp)	End (bp)	Accessibility			Log ₁₀ (Wellington P -value)
			Left 25 bp	Right 25 bp	Footprints	
B.distachyon	93,120	93,133	9.72	7.96	2.71	-33
	121,569	121,595	8.12	5.20	1.38	-66
O.sativa	93,329	93,357	41.88	52.08	25.95	-240
	121,762	121,789	49.54	41.31	25.18	-283
S.italica	93,775	93,857	71.0	82.33	67.48	-0.24*
	121,613	121,634	310.96	146.16	44.90	-255
S.bicolor	98,062	98,083	80.92	102.56	27.86	-415
	126,405	126,426	101.16	82.12	28.67	-403
Z.mays	97,185	97,206	71.08	49.76	26.43	-386
	125,454	125,479	63.28	19.68	33.04	-391

*: Not significant

Reference:

Piper J, et al. Wellington: a novel method for the accurate identification of digital genomic footprints from DNase-seq data. *Nucleic Acids Research*, 2013, 41: e201-e201.

11. line 289: “reglations” should be “regulation”

Response: Sorry for the mistake, we have corrected it in line 300 as “*regulation*”.

12. line 317-320 and 424-426: Due to the low turnover of cp transcripts, most of the transcripts detected in RNA seq are not reflecting the transcriptional status but rather the post-transcriptional RNA processing status – how that is related with DNA-binding proteins may be potentially interesting (one example for a potential connection between DNA-binding and RNA processing may be given in Méteignier, L.V. et al. 2020). I disagree with the pure statement made here that this could reflect regulation in transcription elongation/termination without mentioning RNA processing status (which is what you measure in cpRNA-seq).

Response: Thanks for your valuable suggestion. We have cited this study (Méteignier et al., 2020) and modified the relevant description according to the reviewer's suggestion (Lines 328-331).

“we found that these CPAFPs tended to be distributed in regions where the RNA-seq coverage was altered, suggesting that they may play a role in regulating transcription elongation, termination, or stabilization”.

13. line 436/437: “thus limiting the construction of a regulatory network of the cpDNA”.

Response: Thanks for your advice, it has been updated as *“thus limiting the construction of a regulatory network of the cpDNA”* now.

14. line 438/439: “may also be conserved among species”.

Response: Thanks for your advice, it has been updated as *“may also be conserved among species”* now.

Reviewer #4 (Remarks to the Author):

In this study presented by Liu et al., the authors compared chloroplast ATAC-seq reads generated from a group of diverse tissues and different grass species. A positive association between highly accessible regions and active chloroplast gene expression is confirmed. Furthermore, an association between footprints that imply protein-cpDNA interactions and known transcriptional regulatory elements is presented in this manuscript.

However, there is a potential technical issue, which in my opinion, should be addressed before analyzing these datasets in depth. According to the “Method” section, the chloroplast ATAC-seq reads were recovered from a regular ATAC-seq protocol, in which nuclei were isolated with a FACS machine. It is a clever idea to use chloroplast-derived reads from such an ATAC-seq experiment and to infer cpDNA

“accessibility”. Nonetheless, it is critical to understand the nature of cpDNA co-isolated with nuclei. Regarding this point, I have the following two concerns:

1) Were these chloroplasts intact or damaged (or fragmented) at the time point of starting Tn5 reactions? If they were damaged, to which extent could the cpDNA accessibility be potentially affected, for example, as a result of interactions between cpDNA and other cellular components present in the homogenate?

Response: Thank you for your insightful comments. According to our experimental procedure, when the Tn5 reaction starts, both status of chloroplasts (complete and broken) may exist, with intact chloroplasts likely constituting the majority. Our experimental procedure is briefly described as follows: Firstly, the sample was chopped with 500 μ L chopping buffer to release the nuclei. After staining with DAPI, 100,000 nuclei were sorted into 1.5 ml tubes using flow cytometry. Nuclei were centrifuged and enriched by removing the supernatant, followed by Tn5 tagmentation at 37°C. Since the chopping buffer contains 0.2% detergent Triton X-100, pores may be formed in the nuclei and chloroplast membranes that facilitate the entry of Tn5. Some chloroplasts may be co-purified with nuclei, as plant cells have plastid–nuclear complexes (Selga et al., 2010). Therefore, although this process removes most of the broken chloroplasts, our data likely contains a mixture of both intact and broken chloroplasts (in the small amount of supernatant left). However, even when chloroplasts are broken, some proteins may continue to bind to cpDNA. As many recent techniques, such as native ChIP-seq (Kasinathan et al., 2014) and CUT&Tag (Kaya-Okur et al., 2019), do not require cross-linking when studying DNA–protein interactions, suggesting that many proteins can remain bound to DNA under uncross-linked conditions. In our study, the identified footprints were co-localized with known functional elements, providing support for our results. Furthermore, similar experimental conditions have been used to study mtDNA, suggesting that our approach is feasible. However, we acknowledge that some proteins may have dissociated from cpDNA during this process and their binding signals may not be detected. We have added more detailed description in the methods section (Lines 480-493) and a relevant discussion in Lines 455-466

For the second question, since sample preparation and filtration in our experiments took only 5 min, flow sorting only 2 min, and centrifugation to remove supernatant only 5 min, the experiments could be completed quickly and were less affected by other cellular components.

Reference:

Kasinathan S, et al. High-resolution mapping of transcription factor binding sites on native chromatin. *Nature Methods*, 2014, 11: 203-209.

Kaya-Okur HS, et al. CUT&Tag for efficient epigenomic profiling of small samples and single cells. *Nature Communications*, 2019, 10: 1930.

Selga T, et al. Plastid-nuclear complexes: permanent structures in photosynthesizing of vascular plants. *Environmental and Experimental Biology*, 2010, 8: 85-92.

2) The experimental procedure was designed to harvest nuclei. Is it true that the cpDNA can be treated as a fair presentation of the entire chloroplast population of the tissue?

Response: In this study, we used FACS to harvest nuclei, we acknowledge that the loss of chloroplasts in this process is inevitable. However, the sequencing coverage of cpDNA in different tissues and species was 1000-fold on average in our study, suggesting that there are sufficient data for subsequent analyses and adequately representing the accessibility of cpDNA. We also recognize that the state of chloroplasts can vary between different cells in a tissue and that our method may preferentially capture chloroplasts that form plastid-nuclear complexes, potentially introducing some bias into our results. We have added more detailed description in the methods section (Lines 480-493) and a relevant discussion in Lines 455-466.

Reviewers' comments:

Reviewer #3 (Remarks to the Author):

The authors addressed all major and minor points I have raised.

Reviewer #4 (Remarks to the Author):

Despite the authors' efforts to improve this manuscript, my second concern regarding the potential bias in profiling the chloroplasts harvested alongside nuclei remains unaddressed. I still question whether the datasets presented here truly represent the typical understanding of "chloroplasts." To clarify, an experimental approach could help exclude this potential bias. By conducting ATAC-seq reactions on a few samples of chloroplasts obtained with a standard chloroplast isolation protocol and comparing them with the data obtained from running a nuclei isolation protocol, the authors could better understand and address this issue.

Reviewer #3 (Remarks to the Author):

The authors addressed all major and minor points I have raised.

Response: We thank the reviewer for the helpful comments on our study.

Reviewer #4 (Remarks to the Author):

Despite the authors' efforts to improve this manuscript, my second concern regarding the potential bias in profiling the chloroplasts harvested alongside nuclei remains unaddressed. I still question whether the datasets presented here truly represent the typical understanding of "chloroplasts." To clarify, an experimental approach could help exclude this potential bias. By conducting ATAC-seq reactions on a few samples of chloroplasts obtained with a standard chloroplast isolation protocol and comparing them with the data obtained from running a nuclei isolation protocol, the authors could better understand and address this issue.

Response: As suggested by the reviewer, we consulted ATAC-seq on isolated chloroplasts and observed similar results.

Firstly, we developed a protocol to isolate chloroplasts based on relevant literature. Chloroplasts were isolated from fresh young leaves of rice (variety Zhenshan 97) about two weeks after germination. The leaves were placed in pre-cooled petri dishes with ice-cold chopping buffer and mechanically disrupted using a blade to yield a chloroplast suspension. This suspension was then filtered through a 30 μm mesh to remove large plant debris. The filtrate was centrifuged at 500 g for 10 minutes at 4°C to sediment the chloroplasts. The supernatant was discarded, and the pellet was gently resuspended in 200 μl of Triton X-100-free chopping buffer and carefully layered onto a 40% Percoll solution prepared in the same buffer. The gradient was centrifuged at 5000 g for 25 minutes at 4°C to separate intact chloroplasts, which remained above the Percoll. These chloroplasts were subsequently collected, diluted in 1 ml of Triton X-100-free chopping buffer, and centrifuged at 1000 g for 10 minutes at 4°C. The final chloroplast pellet was resuspended and used for subsequent Tn5 tagmentation.

After obtaining the chloroplast-ATAC-seq (chl-ATAC) data, we mapped the sequencing reads to the cpDNA of *Oryza Sativa* (RefSeq: NC_001320.1) and removed PCR duplicates. Finally, we got 11,696,981 and 14,716,465 deduplicated mapped reads from the two replicates, covering the chloroplast genome at a depth of 13,042-fold and 16,409-fold respectively (Response Table. 1).

Response Table. 1 Mapping statistics of chloroplast-ATAC-seq data

	Raw reads	Mapping rate	Deduplicated mapped reads	Deduplicated mapped reads ratio
Chl-ATAC-Rep1	14,091,204	98.33%	11,696,981	83.01%
Chl-ATAC-Rep2	17,080,088	98.79%	14,716,465	86.16%

To address the reviewer's concern regarding whether our data can “capture truly chloroplast status”, we performed correlation analysis and clustering of the newly generated chl-ATAC data with the previously conventional ATAC-seq data. The results showed that the chl-ATAC datasets clustered together with the leaf samples (Response Fig. 1a), indicating the characteristics of DNA accessibility changes in cpDNA captured by both datasets are consistent, further supporting the conclusions obtained in our manuscript. Meanwhile, to confirm the validity of the main points raised in our manuscript, we conducted the same analysis using the chl-ATAC data as previously performed with conventional ATAC-seq data. Firstly, we quantified the DNA accessibility of cpDNA by 100 bp bins and compared it with the previously conventional ATAC-seq data. The results showed a high correlation between the two datasets, further elucidating the consistency between the two datasets (Response Fig. 1b-c). Additionally, we replicated the analyses depicted in Fig. 2a and Fig. 3a and observed consistent results (Response Fig. 1d-g). This further confirms the correlation between DNA accessibility changes and expression levels, indicating the biological significance of chloroplast DNA accessibility variations. Moreover, the chl-ATAC data showed consistent footprint patterns with the previous data in known functional elements (Response Fig. 1h).

Response Fig. 1: (a-h) **a**, Heatmap of correlations among chl-ATAC data of young leaves and conventional ATAC-seq data from different tissues. We divided the cpDNA into 100 bp bins and quantified the number of Tn5 insertions per bin, and then calculated the Spearman correlation coefficient between different datasets. **b-c**, The correlation between the two replicates of chl-ATAC data for young leaves with conventional ATAC-seq data for young leaves. Each point represents the number of Tn5 insertions in a 100 bp bin. **d-e**, The distributions of accessibility in genic and intergenic regions. The x-axis is average Tn5 cleavages of per region (first sum the Tn5 cleavages for each genic and intergenic regions,

then divided by its length), while y-axis represents the density. **f-g**, The distribution of accessibility of regions with different expression signals in cpDNA. In the heatmap, the columns represent different levels of RNA-seq coverage and the rows represent different levels of Tn5 signal of ATAC-seq. The numbers in the cell represent the proportion of regions with a certain level of expression that have a certain level of accessibility, for example, the upper-right cell indicates that 40% of the top 20% high expression regions have their accessibility at the top 20% level. **h**, Profiles of Tn5 insertions around the two replication initiation sites, oriA, in multiple tissues of rice.

Taken together, while the conventional ATAC method capture a lower proportion reads of chloroplast, the small size and high copy number of the chloroplast enable the obtained data more than 1000-fold coverage of cpDNA, which is sufficient for subsequent analysis. The high correlation observed between chl-ATAC and conventional ATAC-seq data reflects consistent characteristics of the DNA accessibility in the chloroplast genome. Therefore, we believe that the conclusions drawn previously are generalizable and reflect the true state of cpDNA. We hope that the additional experiments and data analyses will address the concerns raised by the reviewer.

Additionally, we have also expanded the discussion on potential concerns regarding the results presented in our study to encourage readers to critically evaluate our findings in the revised manuscript Line 459-468.

REVIEWERS' COMMENTS:

Reviewer #4 (Remarks to the Author):

The authors addressed all my concerns by demonstrating that there was no technical bias in analyses of chloroplast DNA accessibility by using regular nuclear ATAC-seq data. I recommend accepting this manuscript for publication.

Reviewer #4 (Remarks to the Author):

The authors addressed all my concerns by demonstrating that there was no technical bias in analyses of chloroplast DNA accessibility by using regular nuclear ATAC-seq data. I recommend accepting this manuscript for publication.

Response: Thank you for your helpful comments and suggestions on our study.